# Semiconducting Soft Submicron Particles from the Microwave-Driven Polymerization of Diaminomaleonitrile

**DOI:** 10.3390/polym14173460

**Published:** 2022-08-24

**Authors:** Marta Ruiz-Bermejo, Pilar García-Armada, Pilar Valles, José L. de la Fuente

**Affiliations:** 1Departamento de Evolución Molecular, Centro de Astrobiología (INTA-CSIC), Ctra. Torrejón-Ajalvir, km 4, Torrejón de Ardoz, 28850 Madrid, Spain; 2Department of Industrial Chemical Engineering, Escuela Técnica Superior de Ingenieros Industriales, Universidad Politécnica de Madrid, José Gutiérrez Abascal, 2, 28006 Madrid, Spain; 3Instituto Nacional de Técnica Aeroespacial “Esteban Terradas” (INTA), Ctra. Torrejón-Ajalvir, km 4, Torrejón de Ardoz, 28850 Madrid, Spain

**Keywords:** HCN polymers, DAMN, electrochemical properties, soft submicron particles, microwave-driven polymerization

## Abstract

The polymers based on diaminomaleonitrile (DAMN polymers) are a special group within an extensive set of complex substances, namely HCN polymers (DAMN is the formal tetramer of the HCN), which currently present a growing interest in materials science. Recently, the thermal polymerizability of DAMN has been reported, both in an aqueous medium and in bulk, offering the potential for the development of capacitors and biosensors, respectively. In the present work, the polymerization of this plausible prebiotic molecule has been hydrothermally explored using microwave radiation (MWR) via the heating of aqueous DAMN suspensions at 170–190 °C. In this way, polymeric submicron particles derived from DAMN were obtained for the first time. The structural, thermal decomposition, and electrochemical properties were also deeply evaluated. The redox behavior was characterized from DMSO solutions of these highly conjugated macromolecular systems and their potential as semiconductors was described. As a result, new semiconducting polymeric submicron particles were synthetized using a very fast, easy, highly robust, and green-solvent process. These results show a new example of the great potential of the polymerization assisted by MWR associated with the HCN-derived polymers, which has a dual interest both in chemical evolution and as functional materials.

## 1. Introduction

Diaminomaleonitrile, DAMN, is a π-conjugated compound with electron donor (-NH_2_) and electron acceptor (-CN) substituents, and a high electron affinity, which features a strong intramolecular charge transfer interaction. Its high nitrogen content has meant that it has often been used as a versatile precursor for heterocycles, for example, imidazoles, triazoles, porphyrazines, pyrimidines, pyrazines, diazepines, and purines. DAMN can be used as synthetic precursor of fluorescent dyes, jet-printing links, hair dyes, and biologically active compounds such as insecticides and anticancer agents [1]. In addition, DAMN has potential in the area of chemosensors for the sensing of ionic and neutral species because of its ability to act as a building block for well-defined molecular architectures and scaffolds for preorganized arrays of functionality, as it has been reviewed in the literature [2]. Moreover, there is currently a growing interest in the development of macromolecular systems based on DAMN. In this sense, DAMN is the formal tetramer of hydrogen cyanide, HCN, and the potential applications of the DAMN polymers should be understood in the light of the up-to-date attention of the HCN polymers in the field of materials science. The DAMN polymers are a particular group within the more general set of the HCN-derived polymers.

HCN-derived polymers, or simply HCN polymers, can be currently considered as a new promising type of multifunctional soft materials [3,4,5]. This heterogeneous family of complex substances present interesting properties that consequently encourage the development of emergent photocatalyzers [6], semiconductors, nanowires, ferroelectric materials [7], capacitors [8], coatings with potential biomedical applications [9,10,11], protective films against corrosion [12,13] and bacteria [14], antimicrobial media for passive filters [15], and biosensors [16] based on the HCN oligomerization/polymerization chemistry.

HCN polymers have been obtained under multiple conditions, i.e., using net liquid HCN together with a basic catalyzer such as ammonia or trimethylamine [17,18]; by the irradiation of HCN gas or aqueous solutions [19,20,21]; from alkaline water solutions of pure HCN or their soluble salts such as NaCN or KCN (see e.g., [22,23,24,25,26,27,28]); through the wet polymerization of its trimer, aminomalononitrile (AMN), at room temperature [29]; by DAMN, using thermal activation, both in bulk [16,30,31,32] and in an aqueous solution [8]; and even by the heating of its hydrolysis product, formamide [33]. Moreover, the generation of HCN polymers assisted by the microwave radiation (MWR) of formamide [34] and cyanide [35] has been recently described.

The advantages of using MW dielectric heating for performing polymerization reactions are well known (e.g., the remarkable reduction in polymerization time, improved monomer conversion, and clearer reactions than the ones conducted under a conventional thermal system (CTS)). In addition, the production of HCN polymers at the nanoscale was only achieved when this innovative synthetic methodology was applied to aqueous NH_4_CN polymerizations varying the reaction times and the temperatures between 130 and 205 °C [36]. In the present work, the production of soft submicron particles from the MW-driven polymerization of DAMN is described, whereas conventional thermal polymerization conditions were found to be ineffective in this regard [8]. On the other hand, it has been suggested that the integration of polymeric HCN-derived nanoparticles as fillers in different matrixes could lead to the design of novel composite materials [8].

Beyond these morphological aspects, the redox behavior of the several HCN polymers is remarkable. The NH_4_CN polymers can be considered as insolating materials; on the contrary, the DAMN polymers synthetized under hydrothermal conditions present a capacitive performance [8]. Even DAMN polymers produced in bulk (in the absence of a solvent), in both solid and molten state, have shown to be efficient materials for the modification of electrodes and the development of biosensors [16]. Thus, it seems interesting to explore the DAMN polymers for the design of electronic devices and the potential generation of new composites with appropriate electrochemical properties.

While considering the potential of the assisted MW syntheses for the production of polymeric HCN-derived nanoparticles and the singular electrochemical characteristics of the DAMN polymers previously reported; herein, a new generation of submicron-sized highly conjugated polymeric materials are described together with a detailed study of their structural, thermal, and redox behaviors. In concrete, DAMN polymers were synthetized at 170 and 190 °C under the presence of air or an inert atmosphere of nitrogen. Therefore, the results reported in the present work not only can be of interest for the design of coming multifunctional materials, but also in the field of the origins of life, since they could enrich the understanding of prebiotic chemistry [37].

## 2. Materials and Methods

### 2.1. Synthesis of the DAMN Polymers

All the DAMN polymers synthetized in this work were synthetized from aqueous suspensions of 135 mg of commercial DAMN (98%, purchased from Sigma Aldrich (St. Louis, MO, USA) and used as received) in 5 mL of distillated water, under air or N_2_, using 20 mL capacity vials fitted with Teflon/silicone septa. The reaction times, temperatures, and the work pressures for each experiment are indicated in the Table 1. A Biotage Initiator + microwave reactor purchased by Biotage (Sweden) was used in the present study. The power supplied by the reactor is in the range from 0 to 400 W from a magnetron at 2.45 GHz, with a power of about 70–95 W for the temperatures chosen in this work. No additional efforts were made regarding the heating ramp to obtain the desired work temperature, as was the case for the NH_4_CN polymerizations [35]. No pressure peaks were observed for the release of gaseous species such as ammonia. The final dark suspensions were vacuum filtered using glass fiber filters (Merck Millipore Ltd. (Burlington, MA, USA) and washed with water. For details of the MW reactor and of the filtration system, please see Figure 1 in [36]. In addition, Figure 1 shows a graphical diagram for the synthesis of the DAMN polymers described herein. The gel fractions were collected and dried under reduced pressure. Polymeric conversions, α, were calculated as α (%) = [(final weight of insoluble dark polymer/initial weight of DAMN)·100].

### 2.2. Characterization of the DAMN Polymers

#### 2.2.1. Elemental Analysis

Approximately 5 mg of polymer sample was examined for determination of the mass fractions of carbon, hydrogen, and nitrogen by using a PerkinElmer elemental analyzer, model CNHS-2400. The oxygen content of the samples was calculated by means of the difference.

#### 2.2.2. FTIR Spectroscopy

Diffuse-reflectance spectra were acquired in the 4000–400 cm^−1^ spectral region with an FTIR spectrometer (Nicolet, model NEXUS 670) configured with a DRIFT reflectance accessory (Harrick, model Praying Mantis DRP) mounted inside the instrument compartment. The spectra were obtained in CsI pellets, and the spectral resolution was 2 cm^−1^.

#### 2.2.3. C Solid-State Cross Polarization/Magic Angle Spinning Nuclear Magnetic Resonance (CP/MAS) NMR

^13^C CP MAS NMR spectra were obtained with a Bruker Advance 400 spectrometer and a standard cross-polarization pulse sequence. Samples were spun at 10 kHz, and the spectrometer frequency was set to 100.62 MHz. A contact time of 1 ms and a period between successive accumulations of 5 s were used. The number of scans was 5000, and the chemical shift values were referenced to TMS.

#### 2.2.4. Powder XRD

Powder XRD was performed with a Bruker D8 Eco Advance with Cu_K**α**_ radiation (λ = 1.542 Å) and a Lynxeye XE-T linear detector. The X-ray generator was set to an acceleration voltage of 40 kV and a filament emission of 25 mA. Samples were scanned between 5 (2θ) and 50° (2θ) with a step size of 0.05° and count time of 1 s in Bragg–Brentano geometry.

#### 2.2.5. UV-Vis Spectroscopy

UV-Vis spectra were obtained using an Agilent 8453 spectrophotometer. All spectra were recorded in DMSO [38].

#### 2.2.6. Thermal Analysis

Thermogravimetry (TG), differential scanning calorimetry (DSC), and differential thermal analysis (DTA) were performed with a simultaneous thermal analyzer model SDTQ-600/Thermo Star from TA Instruments. Non-isothermal experiments were carried out under dynamic conditions from room temperature to 1000 °C at a heating rate of 10 °C min^−1^ under argon atmosphere. The average sample weight was approximately 20 mg, and the argon flow rate was 100 mL min^−1^. In the present case, only the TGA data were discussed. A coupled TG-mass spectrometer (TG-MS) system equipped with an electron-impact quadrupole mass-selective detector (model Thermostar QMS200 M3) was employed to analyze the main species that evolved during the dynamic thermal decomposition of all of the samples.

#### 2.2.7. Scanning Electron Microscopy (SEM)

The surface morphologies of the polymers were determined by a ThermoScientific Apreo C-LV field emission electron microscope (FE-SEM) equipped with an Aztec Oxford energy-dispersive X-ray microanalysis system (EDX). The samples were coated with 4 nm of chromium using a sputtering Leica EM ACE 600. The images were obtained at 10 kV.

#### 2.2.8. Particle Size Analysis

The values of the Z-average and polydispersity index (PdI) were registered using a Zetasizer Nano instrument (Malvern Instruments Ltd., Almelo, the Netherlands), employing ethanol as a solvent to scatter the samples. Approximately 0.2 mg of the polymeric sample was suspended in 2 mL of ethanol and sonicated for 15 min, and the resulting solution was then measured.

#### 2.2.9. Surface Area

Brunauer–Emmett–Teller (BET) surface areas were evaluated by nitrogen adsorption–desorption isotherms obtained at −196 °C using a Micromeritics ASAP 2010 device. Before each measurement, the samples were degassed at 250 °C for at least 3 h.

#### 2.2.10. Density

Density for DAMN polymers’ data were obtained in a helium pycnometer AccuPyc 1340 of Micromeritics.

#### 2.2.11. Gel-Permeation Chromatography (GPC)

The apparent molecular weights of the DAMN polymers were estimated by gel-permeation chromatography (GPC) using poly(methyl methacrylate) as standard (Polymer Laboratories, Laboratories, Ltd. (USA) ranging from 2.4 × 10^6^ to 9.7 × 10^2^ g/mol) from suspensions of 8 mg/mL of DAMN polymers in dimethylformamide (DMF).

### 2.3. Electrochemical Measurements

An Ecochemie BV Autolab PGSTAT 12 with a conventional three-electrode cell at 20–21 °C was used in the electrochemical studies. The electrodes were a Pt disc (3 mm diameter) as working electrode, a Pt wire as auxiliary, and an Ag/AgCl/KCl 3M (E = 0.205 vs. ENH) as reference electrode. All electrochemical measurements were in dimethyl sulfoxide (DMSO) with NaClO_4_ 0.1 M as supporting electrolyte. All the polymer DMSO solutions were stable and showed no changes even during or after the electrochemical experiments.

## 3. Results and Discussion

### 3.1. Synthetic Conditions and Chemical Composition of the DAMN Polymeric Submicron Particles

Considering the technical specifications of the MW reactor’s manufacture, the reaction times used here in each case are roundly equivalent to a reaction time of 144 h at 80 °C using a CTS. It is known that in the case of cyanide polymers, if equivalent reaction times are used, the main structural characteristics are preserved when the hydrothermal polymerization reactions are carried out using a CTS or under MWR, and only the morphology (shape and size) of the polymeric particles is modified [36]. Observe that the properties of the HCN polymers are greatly affected in the synthetic conditions [3]. Therefore, it is mandatory to check if the MWR could only alter the morphology of the DAMN polymeric particles (as in the case of the cyanide polymers) or if it would also have an influence on the structural characteristics of the macromolecular system. For that, a previously well-studied analogous DAMN polymer synthetized using CTS was used as a reference polymer (RP) for comparative purposes. Note that no submicron particles were observed for this RP synthetized under these particular conditions (0.25 M water suspensions of DAMN heating at 80 °C for 144 h under atmospheric pressure) [8].

Table 1 shows the experimental conditions used for the production of soft submicron particles based on the MW-driven DAMN polymerization, and two working temperatures, 170 and 190 °C, were chosen since the temperature could also have a significant influence on the morphological properties of the DAMN polymers [36]. On the other hand, since the final properties of the materials are directly dependent on their structure, it is also mandatory to have as much control as possible on the experimental conditions of polymerization, including the working atmosphere. It has been shown that the atmosphere seems to have a notable influence on the crystallographic properties of HCN polymers, increasing the crystallinity when the reactions are carried out using an inert atmosphere [35]. Thus, experiments under nitrogen (Table 1, polymers **1** and **2**), or under atmospheric air (Table 1, polymers **3** and **4**) were carried out. In addition, the experiments conducted under anoxic conditions could simulate prebiotic hydrothermal conditions, because DAMN can be considered as an important prebiotic precursor [24].

The conversion (α) obtained for all the reactions were all similar and about 35% was found (Table 1), independently of the pressure atmospheric work (air or nitrogen). Therefore, the hydrothermal polymerizations of DAMN do not seem to be influenced by the presence of air, as was the case of the MW-driven polymerizations of cyanide. In that case, the yield of the reactions was substantially diminished under an inert nitrogen atmosphere [36]. However, the lower conversions reached for polymers **1**–**4** compared to the RP (~75%) seems to indicate that the secondary oxidation processes from DAMN to diiminosuccinonitrile, and the hydrolysis reactions from DAMN to the formation of formamide, glycine, and aminomalonic acid, could be notably increased by the high temperatures and pressures used in the present study, in agreement with a previous proposal made by Ferris and Edelson [39] (Figure 2). Moreover, it is important to mention that we also observed experimentally via gas chromatography-mass spectrometry (GC-MS) the formation of urea, oxalic acid, glycine, and formamide in the DAMN polymerization processes under hydrothermal conditions using a CTS (data not shown). Thus, the minor available amount of DAMN likely leads to a diminution of the precipitated DAMN polymers.

On the other hand, the results of the elemental analysis indicate very similar chemical compositions for the four polymers (Table 1). Moreover, the N/O ratios are comparable to the RP, while the C/N values are slightly greater; however, the C/H data are somewhat lower please see [8]. The roughly empirical formulas for polymers **1**–**4** are C_3_H_3_N_2_O, C_13_H_13_N_10_O_4_, C_15_H_14_N_12_O_5_, C_12_H_12_N_10_O_4_, and respectively; and C_5_H_5_N_4_O_2_ for the RP.

### 3.2. Structural Characterization

The structural characteristics of polymers **1**–**4** inferred by FTIR, ^13^C NMR, and XRD corroborated the high similarity between them and also with the RP [8] (Figure 1). The IR spectra of polymers **1**–**4** show four regions centered at 3343, 2202, 1653, and 640 cm^−1^, which have been largely described elsewhere (see e.g., [38,40,41]). In brief, these bands can be related to amine and hydroxyl groups, nitrile, and imine bonds (Figure 1a). The nitrile bands are more intense than that observed in the equivalent RP sample. This observation can be quantitatively calculated by the EOR values (extension of the reaction, [42]) (Table 2). However, no other significant differences were observed between the FTIR spectra from polymers **1**–**4** and from the RP. The values of the EOC (extension of the conjugation [35]) were comparable for all the polymers (Table 2). Thus, at first glance, this quantitative spectroscopic analysis seems to indicate that the DAMN polymers synthetized using MWR or a CTS present similar macrostructures, only with a slight increase in the EOR value. These differences can also be observed by ^13^C NMR and XRD.

**Figure 1 polymers-14-03460-f001:**
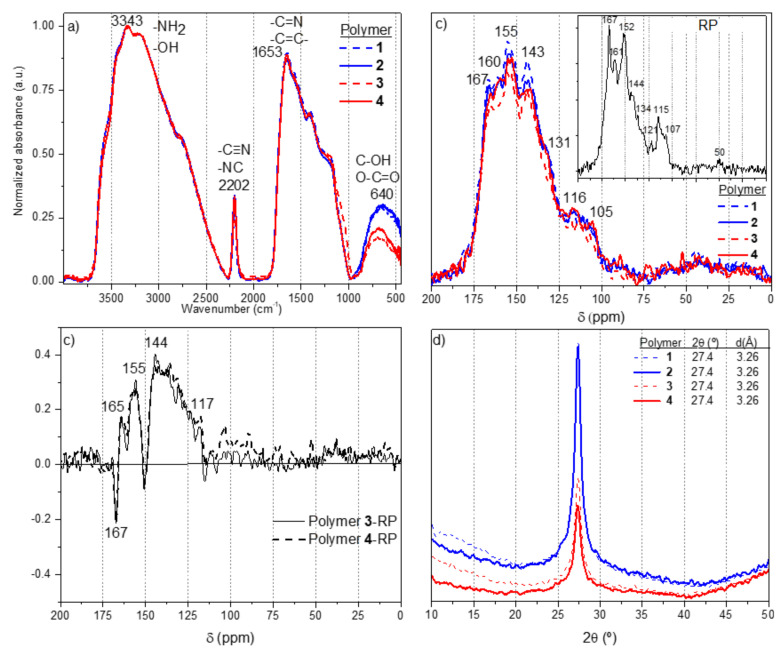
(**a**) Representative FTIR spectra of polymers **1**–**4**. (**b**) Solid state ^13^C NMR spectra of polymers **1**–**4**. In the insert plot, the ^13^C NMR spectrum of the RP is shown [8]; (**c**) Subtractions of the ^13^C NMR spectra; (**d**) Representative XRD patters of polymers **1**–**4**.

The solid-state CP MAS ^13^C NMR spectra for polymers **1**–**4** are very similar between themselves and no notable differences were found, showing broad and unresolved resonances at 167, 160, 155, 143, 131, 116, and 105 ppm (Figure 1b). These same resonances were also observed in the ^13^C NMR spectrum of the analogous RP (insert plot in Figure 1b). These signals can be assigned to carbon in nitrogenous functional groups, such as imines and nitriles, or being part of imidazoles or pyrazines rings; additionally, they can also be assigned to the carbon of the oxygenated groups such as amides and on heterocycles such as oxazoles [8]. The subtractions of these spectra of polymers **3** and **4** to the ^13^C NMR spectrum of the RP are indicated in Figure 1c. The macrostructures of the polymers **3** and **4** present additional peaks at 165, 155, 144, and 117 ppm with respect to the RP. The first one can be attributed to C=O from esters of the amides, the second resonance to –C=N- from azomethine groups and/or from N-heterocycles, the broad signal with a maximum at 144 ppm can be assigned to C=C in heteroaromatic systems, and the last one at 117 ppm with nitrile groups (in good agreement with the FTIR spectra). Thus, it seems that the DAMN polymers from the MWR syntheses present a greater portion of N-heterocycles and aromatic carbons than the RP. It was proposed that the DAMN polymers synthetized under conventional hydrothermal conditions are complex mixtures of extended conjugated systems, lineal or cyclic, based on their NMR spectra [8], as it is shown in Figure 3. The comparison between the FTIR data and the subtractions of the ^13^C NMR spectra appears to indicate that the MWR improves the formation of the N-heterocyclic fractions and the polyamide portion in the structures of the DAMN polymers. The XRD patters of polymers **1**–**4** show only one peak at 2θ = 27.4°, which can be related to the (002) diffraction of the graphitic layers [37], with interlayer distances of 3.26 Å in all the cases (Figure 1d). However, they are considerably different to those found for the RP [8], since the weak peak related to the (100) reflections is missing when the polymers are synthetized by MWR. This result may be related to a different in-plane structural packing motif. In this way, polymers **1**–**4** could present amorphous, disordered, in-plane and out-plane structures with only a nanoscopic range order by the stacking of the main structural motif due to the lack of peaks at lower 2θ [43,44]. On the other hand, when the polymerization of DAMN is carried out under a nitrogen atmosphere, the intensity of the peaks is increased while their width is decreased. This fact probably points out a higher order in the stacking of the two-dimensional (2-D) macrostructures for polymers **1** and **2** (Figure 1d), in agreement with previous results [35] as commented above.

Finally, the UV-Vis spectra of polymers **1**–**4** show a main absorption at around 430 nm, which may be related to a π–π * transition along the polymeric backbone (Figure 2) [45]. Moreover, the samples synthetized at 170 °C (polymers **1** and **3**) exhibit an additional weak band centered at ~310 nm, which is likely due to aromatic fragments that have a naphthalene-like frame [46].

### 3.3. Thermal Stability

The derivative thermogravimetry (DTG) curves can be considered as excellent fingerprints for HCN-derived polymers for samples showing similar spectroscopic data, but synthetized under different experimental conditions; thus, the derived fragments from the thermogravimetry-mass spectrometry (TG-MS) analysis may be very useful for the structural elucidation of these types of complex systems [47]. In Figure 3, the DTG-grams and the TG-MS curves of polymers **1**–**4** are shown. For all the samples, three thermal degradation steps are observed (drying, <225 °C; main decomposition, 225–450 °C; carbonization, >450 °C) (Figure 3a,b), displaying temperatures of the DTG maxima around 80–90, 240, 420, 680, and 830 °C, and leaving char residues of ~25% in weight, in agreement with previous results from the RP [8], with the exception of a DTG peak around 330 °C, which is not found in the polymers synthetized herein. These results denote a thermo-structural resemblance, but no identical structures for the DAMN polymers synthetized under hydrothermal conditions using MWR against a CTS, according to the spectroscopic data discussed above. The lack of the thermolabile groups at ~330 °C could be related to the greater proportion of structures based on N-heterocycles and could therefore be more thermally stable than the corresponding lineal blocks or segments on the polymeric backbone found in the DAMN polymers synthetized using MWR, as commented above.

Similar conclusions can be inferred from the TG-MS traces. Polymers **1**–**4** show the same fragmentation pattern (Figure 3c–f). The fragments at m/z = 16, 17, 18, 26, 27, 28, 43, 44, 45, and 52 can be assigned to ^+^NH_2_, ^+^OH/^+^NH_3_, ^+^H_2_O, ^+^CN, ^+^HCN, ^+^N_2_, ^+^H_3_CN_2_, formamidine ^+^H(C=NH)NH_2_/^+^CO_2_, formamide ^+^HCONH_2_, and ^+^-CH=CH-CN, respectively. All these signals were also identified previously in other HCN polymers [16,47], but the profiles of the curves are different depending on the experimental synthetic conditions used for the HCN polymers’ production. This once again reveals the structural similarity in the macromolecular backbone of all the HCN polymers with the main carbonization process associated with the decyanation of the structure (*m*/*z* = 26 and 27, ^+^CN and ^+^HCN, respectively) and related to a structure based mainly on a conjugated –C=N- system. In addition, considering the oxygenated fragments (*m*/*z* = 44 and 45, ^+^CO_2_ and formamide, respectively) and the shape of the corresponding curves suggests that there are at least two classes of oxygenated functional groups. On the one hand, the weaker bonds are present in the pendant groups (for example, -COOH or -CONH_2_ from the hydrolysis of –CN groups) or from the generation of lineal polyamides and/or polyacids (<450 °C); on the other hand, the strongest bonds are present in the macrostructure formed, for instance, by oxazole rings (>450 °C) (Figure 3). However, in any case, the presence of oxygenated functional groups only can be related to the role of the water in the pathways of the formation of the macrostructures, being independent of the occurrence of the air in the reaction environment. In addition, all these fragments can be correlated with the macrostructures proposed in Figure 3 [47].

### 3.4. Morphological and Textural Properties

As it was indicated above, the MWR entails two important modifications both in the shape and the size of the HCN-derived polymeric particles [35,36]. Thus, the RP presents particles with no well-defined morphology at the micrometric scale. However, polymers **1**–**4** are much smaller and present uniform shapes, especially the polymers **1** and **3** (Figure 4). In polymers **2** and **4**, a few long, isolated nanofibers were observed together with particles with a long rice-like shape similar to those previously described for the MW-driven polymerization of cyanide [35]. However, for the first time, it was found that the DAMN polymeric submicron particles from polymers **1** and **3** exhibit only one pure morphology. Previously, only mixtures of different morphologies and sizes for insoluble HCN-derived polymeric particles were described, from microspheres to laminated particles and with no well-defined shape microparticles [8,35,36].

In addition, the size distributions of polymers **1**–**4** present a Z-average of ~230 nm (Figure 5, Table 3), with a narrower distribution for the samples prepared at 190 °C (polymers **2** and **4**), despite the presence of single isolated nanofibers. On the other hand, it was statistically proven that the molecular sizes for the cyanide polymers were inversely related on the O content present on the macrostructure when the reactions were conducted by MWR [36]. However, as it can be seen in the insert plot in Figure 5, for the DAMN polymers, there seems to be a contrary behavior, i.e., a higher O percentage leads to larger particles. Again, it is demonstrated that the properties of the HCN polymers are sensitive to the experimental conditions chosen for their production, in this case, the reagent used for their syntheses. Moreover, the size distributions seem to depend on a greater grade of the polymerization temperature than the reaction atmosphere (air or N_2_), which is in good agreement with the independence of the O proportion due to the presence or lack of air during the polymerization processes.

On the other hand, other textural and physical properties were measured (Table 3). The BET surface area of polymers **1**–**4** have values around 30 m^2^/g. Moreover, the density values of polymers **1**–**4** are roughly 2.2 g/cm^3^, being higher than the only value reported in the bibliography for a HCN polymer, 1.62 g/cm^3^, which was obtained during the manufacturing process of the HCN in the DuPont gas-manufacturing system [41]. Finally, the number-average molecular weights (M_n_) obtained by the GPC analysis for all the samples were determined. These apparent molecular weights must be taken with care because the solubility of the DAMN polymers is limited in DMF, and it is difficult to have a similar standard due to the high complexity of these systems, as it was shown in Figure 3. In any case, a high polymerization temperature could increase the M_n_ values, at least in those samples prepared under an N_2_ atmosphere.

As a summary of all these multi-technical characterizations for polymers **1**–**4**, it can be said that the DAMN polymerization assisted by MWR leads to the generation of complex mixtures of lineal and cyclic structures based mainly on C=N bonds (Figure 3) with an apparent greater proportion of N-heterocycles and amide groups with respect to the analogous polymer synthetized using a CTS. At least a fraction of these heterogeneous polymeric systems are 2-D macrostructures that are able to form stackings. This layer arrangement is influenced by the use of an inert atmosphere during the polymerization processes. Moreover, the MWR has allowed for the preparation, for the first time, of uniform DAMN polymeric submicron particles with surface areas around 30 m^2^/g and a density of ~2.2 g/cm^3^.

The close resemblance of the chemical and structural characteristics of polymers **1**–**4** is also reflected in their electrochemical properties, showing a similar redox behavior, which will be discussed in the next section. Noteworthily, the decrease in the particle size of the DAMN polymers described herein has enabled the measurement of their electrochemical properties in DMSO solutions against the voltammetry measurements made using pressing powders of the RP [8].

### 3.5. Electrochemistry

In order to complete the characterization of the new DAMN polymers, their electrochemical properties were studied. All the polymers showed similar cyclic voltammograms (CVs) with two stable redox systems, a quasi-reversible one at around −0.7 V, and an irreversible one at 0.4 V approximately. Figure 6 shows the CV of polymer **1** as an example.

In order to determine the electrochemical properties of the four polymers, the reversible redox system has been chosen for a detailed study. Figure 7 shows the cyclic voltammograms of about 3 × 10^−5^ M solutions of polymers **1**–**4** in NaClO_4_/DMSO. As it can be seen, all the polymers present similar CVs, with a quasi-reversible electrochemical system at very close potentials, corresponding to the same electroactive functional groups, which is in agreement with their very similar macrostructures. The polymers’ corresponding Tafel plots (overpotential vs. log i) [48] are shown in Figure 8. The slopes of the Tafel plots allowed us to calculate the electron transfer coefficient, α, the number of electrons exchanged, n, and the exchange current density, *j*_0_. The results shown in Table 4 indicate that, in all cases, the corresponding redox systems exchange one electron, and regarding the α values, we can conclude that their symmetry is similar. That is to say, this electrochemical system is the same in the four DAMN polymers.

In addition, the exchange current density, *j*_0_, equal to *j_c_* or *j_a_* at equilibrium (*E = E*^0^) when the net current is zero, and indicative of the reversibility of the redox systems, allowed us to estimate the standard rate constant of the electrochemical reaction, *k*_0_, from the *j*_0_ = *n F k*_0_
*C* equation. The obtained values are very close, confirming the similar electrochemical behavior of the four solved polymers.

On the other hand, the peak current for a reversible system (at 25 °C) follows the Randles–Sevcik equation [49]:*i_p_* = (2.69 × 10^5^) *n*^3/2^
*A C D*^1/2^
*v*^1/2^
where *n* is the exchanged number of electrons, A is the electrode area, *C* is the concentration (in mol/cm^3^), *D* is the diffusion coefficient, and *v* is the potential scan rate. That is to say, the current must be directly proportional to the concentration and increases with the square root of the scan rate. Thus, the dependence on the scan rate is indicative of an electrode reaction controlled by mass transport. Figure 9 shows the obtained plots of peak current density, normalized with the corresponding concentrations, vs. the square root of the scan rate. All polymers show a linear relationship with the scan rate and the slopes allow us to calculate the diffusion coefficients. The slopes of the lineal fits in Figure 9 suggest that the diffusion coefficients are different and related to the different molecular weights of the polymers. These results also indicate the electrochemical stability of the polymers.

In addition, the study of the diffusion with rotating electrode by the Koutecky–Levich treatment [48] allows us to determine the influence of the kinetics in the global process and the value of the rate constant of the forward reaction (reduction) at the measurement potential, and to calculate the diffusion coefficient. The limiting current for the voltametric wave is expressed by the Koutecky–Levich equation:$$\frac{1}{i_{l}}=\frac{1}{nFAk_{f}C}+\frac{1}{0.62nFAD^{2/3}\nu^{-1/6}C\omega^{1/2}}$$
where *n* is the number of electrons, *F* the Faraday constant, *A* the electrode area, *ν* the DMSO kinematic viscosity (0.0214 cm^2^·s^−1^), *D* the diffusion coefficient, *ω* the angular velocity, *k_f_* the heterogeneous second-order rate constant, and *C* the bulk polymer concentration (in mol cm^−3^). The slopes of the plots 1/*i_l_* vs. l/*ω* ^l/2^ allow us to calculate the diffusion coefficients and the intercepts give us the rate constants. Figure 10 shows the Koutecky–Levich plots obtained with the four polymers at *E* = −0.8 V. The obtained diffusion coefficients are close to those obtained by the Randles–Sevcik equation, and the mean values are collected in Table 4. The values are in concordance with the polymers’ size (see data of the Table 3). On the other hand, the obtained *k_f_* values are very close and notably higher than the standard value *k*_0_, indicating a very fast electrochemical reaction at *E* = −0.8 V. 

Finally, in order to complete the electrochemical study and consider the shape of the CVs (Figure 6), we have considered it to be of interest to study the possibility of applying these materials as semiconductors. The formal potentials of the two systems showed in the CVs can be related to the p- and n-doping status of a semiconductor material and can be used to calculate the HOMO (highest occupied molecular orbital), the LUMO (lowest unoccupied molecular orbital), the band gap (the gap between the highest occupied molecular orbital (HOMO)), and the lowest unoccupied molecular orbital (LUMO)) [50,51]. For these calculations, the values of the absolute potential for the standard hydrogen electrode (SHE) and the potential difference between the Ag/AgCl and SHE of 4.43 eV and 0.22 V were used. The formal potentials are usually taken as the average of the anodic and cathodic peak potentials for both p- and n-doping systems. As expected, all the polymers showed the same values for the formal potentials. Table 5 collects the results obtained, which demonstrates that the synthesized polymers could be applied as good semiconductors.

In this work, the electrochemical characterization of the new polymers has been carried out. The preparation of the modified electrodes and their application for the development of electrochemical devices is beyond our aim at this moment. In view of the results obtained in the electrochemical characterization shown in this work, it is possible that we will address the studies of modified electrodes in further work.

### 3.6. New Insights in Prebiotic Chemistry and Materials Science

Classically, HCN polymers have been considered preferential prebiotic precursors of biomonomers [52] and the DAMN as a key intermediate in the oligomerization/polymerization of HCN [3,24,39]. Beyond the interest in the generation of organics with biological applications, it was shown herein that DAMN under the simulated conditions of plausible prebiotic-subaerial-hydrothermal environments (high temperatures, relatively high pressure, and a lack of oxygen) lead to the generation of semiconducting submicron particles, which could notably enrich the chemical space and enable the discovery of new protobiological reaction networks due to their potential as catalyzers and photocatalyzers. These results are especially interesting when considering the hypothesis about the role of hydrothermal scenarios as good niches for the origins of life or at least for the advancement of molecular complexity [53,54]. On the other hand, the robustness of the possible prebiotic syntheses of HCN polymers offers additional competitive environmental and economic advantages for the development of a new class of multifunctional polymeric materials using low-cost, easy-to-produce, and green-solvent processes. Moreover, the DAMN polymerization is not affected by the presence of air, thereby improving the handle of this one-pot process. On the contrary, it was also shown that the DAMN is highly sensitive to hydrolysis and oxidation reactions (Figure 1). This fact leads to the decrease in the available amount of DAMN susceptible to polymerization. A further research work is in progress to solve this weakness of the hydrothermal polymerization of DAMN, using solvents other than water. The preliminary results indicate that it is possible to obtain conversions for DAMN polymers around ~90% with a narrower particle size distribution than reported herein. In any case, this research showed the high capacity of the MWR for reducing the reaction time of polymerization to a few minutes, driving the formation of DAMN soft submicron particles as semiconductors as a first step in the development of fillers in new composite materials and others important applications such as medicine [55]. In addition, the results discussed in the present work complete and extend the knowledge about related π-conjugated systems based on diaminoanthraquinones [56], oligothiophenes with dicyanomethylene groups [57], and C=N based systems with donor-acceptor intramolecular charge transfer [58] with optoelectronic, amphoteric redox and semiconductive properties, respectively.

## 4. Conclusions

Bearing in mind that the possibility of tuning the morphology of HCN polymers synthesized through MWR has been recently discovered, in the present study, DAMN polymerization was explored with this technique. To achieve this, DAMN aqueous suspensions were subjected to MWR at 170 and 190 °C and short reaction times, 3 and 16 min, respectively. Although moderate yields of insoluble black polymers were found, their textural and morphological characteristics at the submicron scale allowed us to approach the study, for the first time, of their redox properties in solution. Thus, the cyclic voltammetry of these highly conjugated polymeric submicron particulate systems in DMSO solution provided relevant information concerning their electrochemical behavior. In addition, the diffusion coefficients and different kinetic parameters were estimated and confirmed the high similarity of the four samples produced in this work, agreeing with the conclusions reached after the complete structural and thermal analyses were carried out. Moreover, the new submicron particles synthetized from the air tolerant and very fast MW-driven polymerization of DAMN can be considered as semiconductors based on the values of the band gaps found. This study presents novel results that on the one hand increase the prebiotic chemical space to be explored, and on the other hand show significant improvement in the production of the HCN polymers. As a general conclusion, DAMN is a π-conjugated compound with a great potential for developing novel soft materials in modern polymers science.

## Data Availability

Not applicable.

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
