# Peer review of "Semiconducting Soft Submicron Particles from the Microwave-Driven Polymerization of Diaminomaleonitrile"

_polymers, 2022, doi:10.3390/polym14173460_

Round 1
Reviewer 1 Report
The authors presented a robust and quick synthetic method for making diaminomaleonitrile-derived polymers using microwave radiation. A series of four polymers were synthesized under this technique. The analytical characterization and data interpretation were thorough and carefully considered. Generally, the manuscript is well-organized and presented in a logical manner. I suggest it for publication in Polymers after addressing the following issues.
1). The technical term "HCN" appeared multiple times in the paper. The authors need to provide the full name for the general readership. Otherwise, a non-professional may not know what "HCN" stands for.
2). The method description, in particular, the instrumentation, in Section 2.2 is not complete. More details should be made available for readers without going back and searching for another reference article.
3). The authors may want to move the denotation (a) and (b) in Table 1 (Page 4, Line 159 - 163) right under the table to make the title cleaner and easier to read.
4). How is the oxygen content measured in Figure 5? What are the concentrations of the polymeric nanoparticles for DLS measurements?
5). The Discussion section can be merged with the previous Results section.
Author Response
Thank you very much for your comment.
Please, see below our point-by-point response.
1) The technical term "HCN" appeared multiple times in the paper. The authors need to provide the full name for the general readership. Otherwise, a non-professional may not know what "HCN" stands for
- Now in the text is indicated: […] DAMN is the formal tetramer of the hydrogen cyanide, HCN, and […].
2) The method description, in particular, the instrumentation, in Section 2.2 is not complete. More details should be made available for readers without going back and searching for another reference article.
- In the revised version of the manuscript this section has been described in detail.
3) The authors may want to move the denotation (a) and (b) in Table 1 (Page 4, Line 159 - 163) right under the table to make the title cleaner and easier to read.
- We have moved (a) and (b) under the table.
4) How is the oxygen content measured in Figure 5? What are the concentrations of the polymeric nanoparticles for DLS measurements?
- The values of the O % showed in the Figure 5 were calculated by difference from the results of the HCN elemental analysis as it was indicated in (b) in the Table 1 and as it is explained now in the section 2.2.1.
- The preparation of the sample for their DLS measurements is now explained in the section 2.2.8.
5) The Discussion section can be merged with the previous Results section.
- In the revised version only one 3. Results and Discussion section is considered and a new section 3.6. New insights in prebiotic chemistry and materials science has been included.

Reviewer 2 Report
The work presents a comprehensive analysis of novel polymeric nanoparticles derived from DAMN polymers, including also electrochemical measurements. The work is well written and organized and it is of interest to the readers of the journal. Minor revision is requested.
The authors should further examine and discuss the stability of DAMN polymeric nanoparticles for their use in electrochemical systems.
Author Response
Thank ypu very much for your comments.
Please, see below our point-by-point responses.
1) The authors should further examine and discuss the stability of DAMN polymeric nanoparticles for their use in electrochemical systems.
- The thermal and electrochemical stability of DAMN polymeric particles have been studied and described in the manuscript. However, to further clarify this item and in accordance with the reviewer's instructions, we have introduced explicit references to the stability of polymer solutions in DMSO (section 2.3. Electrochemical measurements) and their electrochemical stability (pages 13 and 15, lines 469 and 353 respectively). In addition, as the preparation of modified electrodes and their application for the development of electrochemical devices is out of our aim in this work, we have included a new paragraph, at the end of the section 3.5. Electrochemistry (in page 16, line 582), excluding these studies from the present manuscript.

Reviewer 3 Report
The article entitled "Semiconducting soft nanoparticles from the microwave-driven polymerization of diaminomaleonitrile" by Ruiz-Bermezo et al. represents a new method to prepare nanoparticles. The findings are new and modification of earlier works from the authors. I suggest acceptance of this manuscript after major revision.
The DLS characterization shows clearly the huge distribution of sizes of the DAMN based polymers or oligomers. The authors are suggested to isolate only the polymers (<200 nm size) and characterize them separately as in the title of the paper polymer and nanoparticle does not complement each other. The authors are requested to provide explanation and justification of the yield of nanoparticle formation based on wide DLS spectra obtained.
Many typo should be corrected and sentences should be rephrased. e.g.,
Line 205 can be corrected.
There are some references to be cited:
For similar N-atom and quinone based π-conjugated systems the following citations have to be added.
1. Ganesh M. et al. ACS Omega 2022, 7, 29, 25874-25880. (amino anthraquinones)
2. Aso Y. et al. J. Am. Chem. Soc. 2005, 127, 25, 8928–8929 (importance of conjugated multi-cyano compounds)
3. Zade S. S. et al. Polym. Chem., 2012,3, 1453-1460 (for Donor-acceptor intramolecular charge transfer, C=N based system)
Author Response
Thank you very much for your comments.
Please, see below our responses point-by-point.
1) The DLS characterization shows clearly the huge distribution of sizes of the DAMN based polymers or oligomers. The authors are suggested to isolate only the polymers (<200 nm size) and characterize them separately as in the title of the paper polymer and nanoparticle does not complement each other. The authors are requested to provide explanation and justification of the yield of nanoparticle formation based on wide DLS spectra obtained.
-We agree with the reviewer. In fact, we have used the term “nanoparticle” vs the “microparticles” obtained previously used conventional thermal heating, to highlight that the MWR can lead to the formation of smaller particles. However, in agreement with the referee, really, in the present manuscript we study submicron particles. We have changed the title and correct the main text, indicating submicron particles instead nanoparticles. Taking into account the size distribution showed in the Figure 5, the polymers 2, 3 and 4 showed a size under the micron. As, it is now indicated in the new section, 3.6. New insights in prebiotic chemistry and materials science, a further works is in progress to obtain DAMN polymers with a higher yields and uniform smaller particles, as you can see in the following SEM imagine (preliminary results):
(See word file)
This issue will be addressed in future works and the properties of these new nanoparticles will be studied.
2) Many typo should be corrected and sentences should be rephrased. e.g., Line 205 can be corrected.
-The manuscript has been carefully revised.
3) There are some references to be cited:
For similar N-atom and quinone based π-conjugated systems the following citations have to be added.
- Ganesh M. et al. ACS Omega 2022, 7, 29, 25874-25880. (amino anthraquinones)
- Aso Y. et al. J. Am. Chem. Soc. 2005, 127, 25, 8928–8929 (importance of conjugated multi-cyano compounds)
- Zade S. S. et al. Polym. Chem., 2012,3, 1453-1460 (for Donor-acceptor intramolecular charge transfer, C=N based system)
- These referenced have been added.

Round 2
Reviewer 3 Report
The authors have incorporated the suggested corrections in the revised version of the manuscript. So, I feel the manuscript can be accepted now in its current form.